# Table Foundation Models: on knowledge pre-training for tabular learning

**Myung Jun Kim**                                    *myung.kim@inria.fr*
*SODA Team, Inria Saclay*

**Félix Lefebvre**                                    *felix.lefebvre@inria.fr*
*SODA Team, Inria Saclay*

**Gaëtan Brison**                                    *gb2764@nyu.edu*
*Hi! PARIS, Institut Polytechnique de Paris*
*New York University*

**Alexandre Perez-Lebel**                            *alex@fundamental.tech*
*SODA Team, Inria Saclay*
*Fundamental Technologies*

**Gaël Varoquaux**                                    *gael.varoquaux@inria.fr*
*SODA Team, Inria Saclay*
*Probabl.ai*

**Reviewed on OpenReview:** <https://openreview.net/forum?id=QV4P8Csw17>

## Abstract

Table foundation models bring high hopes to data science: pre-trained on tabular data to embark knowledge or priors, they should facilitate downstream tasks on tables. One specific challenge is that of data semantics: numerical entries take their meaning from context, *e.g.,* column name. Pre-trained neural networks that jointly model column names and table entries have recently boosted prediction accuracy. While these models outline the promises of world knowledge to interpret table values, they lack the convenience of popular foundation models in text or vision. Indeed, they must be fine-tuned to bring benefits, come with sizeable computation costs, and cannot easily be reused or combined with other architectures. Here we introduce TARTE, a foundation model that transforms tables to knowledge-enhanced vector representations using the string to capture semantics. Pre-trained on large relational data, TARTE yields representations that facilitate subsequent learning with little additional cost. These representations can be fine-tuned or combined with other learners, giving models that push the state-of-the-art prediction performance and improve the prediction/computation performance trade-off. Specialized to a task or a domain, TARTE gives domain-specific representations that facilitate further learning. Our study demonstrates an effective approach to knowledge pre-training for tabular learning.

## 1 Introduction: promising, but limited, foundation models for tabular learning

Tables, that often contain an organization's precious data, come with specific challenges to machine learning, as they contain columns of different types and nature. Until recently, deep learning brought little benefits over tree-based models for typical tables (Shwartz-Ziv & Armon, 2022; Grinsztajn et al., 2022; McElfresh et al., 2024). This is in contrast with images, signals, or text, where the latest advances are driven by

foundation models: neural networks pre-trained on a large amount of background data that can be adapted and reused for a great variety of tasks (Bommasani et al., 2021). Pivotal to this vision was BERT (Devlin et al., 2019), showing that repurposing transformer-based backbones (Vaswani et al., 2017) could boost many natural language tasks. Early signs of similar breakthroughs are visible for tabular data. Using pre-trained transformers on synthetic numerical tables, TabPFN outperforms traditional approaches for classification and regression, and can perform data generation and density estimation (Hollmann et al., 2023; 2025).

Beyond modeling numbers, another important challenge of tabular learning is data semantics. Human beings use string in table entries or column names to understand the table. Progress in table *understanding* models leverages these strings (Zhang et al., 2024). But for most tabular *learning* models, including the TabPFN family, the strings are challenging. Such models operate on numbers and they demand that the data scientist convert all columns to numerical representations, a crucial and often tedious operation. Rather, strings can be an opportunity to bring world knowledge in pre-trained models for tables (Kim et al., 2024; Yang et al., 2024). But such existing pre-trained models cannot easily be reused and come with large operational costs, as they require costly fine-tuning to outperform tree-based models.

Indeed, applications must navigate trade-offs between prediction accuracy and operational costs (Bernardi et al., 2019; Paleyes et al., 2022). One benefit of deep learning in vision or text has been to reduce operational complexity via model reuse across tasks (Zhai et al., 2019). Mature foundation models have pushed much further this reuse and operational convenience, as they can easily be specialized with little downstream data (Bommasani et al., 2021). However, their computational cost –for pre-training but also inference– is a real concern (Varoquaux et al., 2024), as illustrated by the stock-market turmoil (Saul, 2025) created by the announcement of the efficient DeepSeek-R1 model (Guo et al., 2025). Table foundation models hold the same promise and face the same peril. Capturing strings and column names enables them to model multiple tables without matching columns (Kim et al., 2024; Yang et al., 2024). This potential is compromised by their compute cost: Kim et al. (2024) report times $200\times$ slower than XGBoost, a very strong baseline. Ideally, repurposing a foundation model should be less, not more, costly than starting from scratch.

Here we introduce TARTE (Transformer Augmented Representation of Table Entries), a pre-trained tabular model. TARTE uses knowledge pre-training to capture associations between strings and numbers. Corresponding representations facilitate downstream learning, fine-tuned or reused as such in combination with other models. In both settings, it gives predictors that outperform the best baselines, based on trees or neural networks. Like recent tabular models, TARTE builds on the playbook of foundation models: broad pre-training to bake in implicit priors that help for a wide variety of tasks. But TARTE takes it much further: we show that its knowledge pre-training does capture information easy to reuse, unlike prior models of strings and numbers where the benefits come from fine-tuning rather than pre-training.

In section 2, we analyze the challenges that tabular data pose to foundation models, and the progress overcoming them across time. We present the TARTE model in section 3: the architecture, a transformer that models string or numerical entries, enriched by column names; the pre-training, on a large knowledge base enriched with numerical attributes from Wikidata; different post-training options, fine-tuned or frozen, combined with another models. Section 4 gives an extensive empirical study. We first show how, given a downstream table, various post-training strategies of TARTE improve the state of the art in tabular learning for different trade-offs between prediction accuracy and computational cost. We study both a small-sample regime, from $n = 32$ to $1\,024$, and mid-sized tables, $n = 10\,000$. We then study the factors of success of knowledge pre-training, showing that it needs diverse and rich pre-training data and works best on complex tables with strings similar to the pre-training data. Finally, we show that TARTE can also be specialized to a domain, enabling a form of transfer learning. Section 5 ends with a discussion and conclusion.

## 2 The unfolding of tabular foundation models

Compared to neural networks, tree-based models have a lead start for tabular learning: their inductive biases match well the properties of tabular data (Grinsztajn et al., 2022). Progress in dedicated neural architectures has recently been closing the gap for large-enough datasets (Borisov et al., 2022; Ye et al., 2024). But tabular foundation models bring the promise of benefits for data of small to moderate size. van Breugel & van der Schaar (2024) argue they should be a research priority, calling for developing properties important to tabular

applications such as cross-dataset modeling as well as handling tables with different types, *e.g.,* numbers and strings. Recent progress on this agenda has required overcoming many table-specific challenges.

**PFNs: learning priors for numerical tables**    TabPFN (Hollmann et al., 2023) brings to tabular learning key ingredients of the success of foundation models: modeling context with transformers. The PFN (prior fitted network) is pre-trained over many datasets chosen to match the domain of interest, here tabular learning. The prediction is "in context": the training set is given as context in the transformer, which uses it to complete the query in a forward pass. As pre-training requires a huge amount of datasets, these must be synthetic, computed with sophisticated random processes. Improving architecture and data generation, the followup TabPFNv2 (Hollmann et al., 2025), and related works (Qu et al., 2025; Liu & Ye, 2025; den Breejen & Yun, 2024), leads to reliably outperforming tree-based models on purely numerical tables. Much ongoing work improves this type of approach, *e.g.,* with better post-training to specialize a model on downstream data (Thomas et al., 2024; Feuer et al., 2024; Koshil et al., 2024; den Breejen et al., 2024, etc).

**Modeling varying schemas and data semantics**    The agenda of table foundation models implies learning across tables and model reuse, and as a consequence modeling tables with different "schemas", different columns with different information. Such setting breaks traditional machine-learning models used on tables, which need correspondence in columns to form features. Given tables with varying number of columns, pre-trained transformers are useful again (even without the in-context learning of PFNs) to construct joint representations from a varying number of inputs (Zhu et al., 2023; Chen et al., 2024). Going further, models should ideally use the data semantics of columns. For instance, two columns may contain numbers, but one being age and the other weight. Capturing these semantics is important, if only to bridge related information across tables. Column names help. A variety of transformer-based models have tackled learning across tables with different columns by adding the columns names to the data used as input: Wang & Sun (2022) learn and transfer across various clinical-trial datasets (and followups, Yang et al., 2024; Spinaci et al., 2024). But, without broad pre-training, transformer-based models do not outperform tree-based models in general.

**Language models as tabular learners**    Large language models (LLMs) are the epitome of models capturing much via broad pre-training, including world knowledge. They work with free-flowing text, not constrained by a schema, and understand the corresponding semantics. They can be adapted to tables *e.g.,* by turning rows into sentences (Dinh et al., 2022; Hegselmann et al., 2023). Dedicated fine-tuning turns LLMs to tabular learners, best performers in very few shot settings (Wang et al., 2023; Gardner et al., 2024). LLMs-based approaches on tables have met more success for tasks beyond tabular learning, such as table understanding, question answering, recognizing columns or entries (Herzig et al., 2021; Zhang et al., 2024).

**Pre-trained models of text and numbers**    The road to build foundation models for tabular learning has stretched between focusing on numbers, as the TabPFN literature, and adapting language models, that deal naturally with the strings in the tables. A body of work shows that tabular learning needs modeling strings but also numbers as such, rather than relying on the tokenization of LLMs. Yan et al. (2024) integrates an LLM but retrain on tables with discretized numerical features. Further from LLMs, Yang et al. (2024) use the column name, cell value, and data type as inputs to a transformer, pre-trained over many tables. CARTE (Kim et al., 2024) reliably outperforms tree-based models on small datasets by pre-training on relational data a graph transformer with an attention mechanism that combines the column name with string encodings or numbers. These models, however, rely on fine-tuning and incur large computational costs.

**Reusing table foundation models**    Language or vision models such as BERT (Devlin et al., 2019), CLIP (Radford et al., 2021), or the many followups ended up being called "foundation model" because their ease of reuse and specialization to many application-specific models. The huge popularity of specializing models is visible for instance on the huggingface hub, that hosts more than a million models, many of which are derived by reusing already pre-trained models, to offset pre-training costs. LLMs can even give backbones for table predictors, as mentioned above. Reuse and transfer is also high stakes for tabular data (Levin et al., 2023), and here modeling across tables with unmatched columns is particularly useful, to bring in data with different schema. This ease of specialization and reuse has not really been demonstrated for tabular data. Maybe the closest result comes from CARTE (Kim et al., 2024) which demonstrates benefits of specializing to

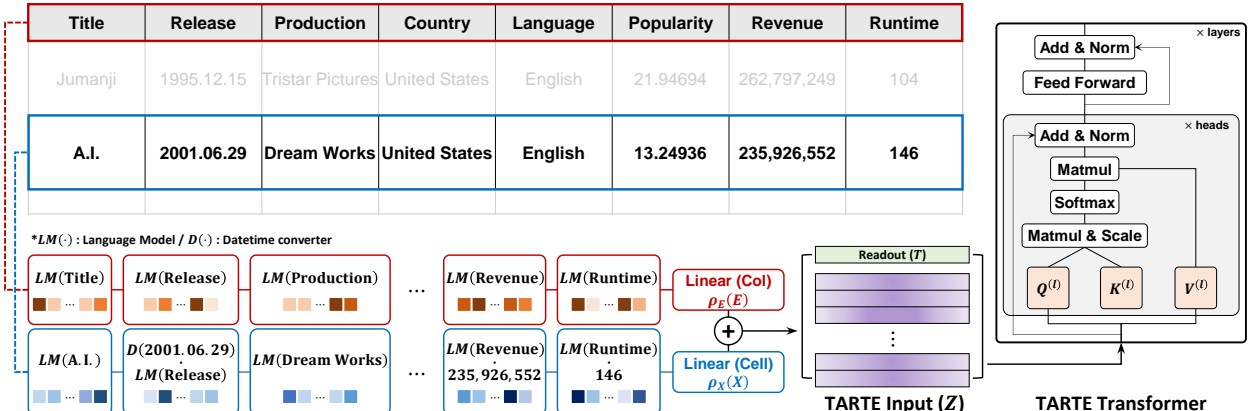

Figure 1: **Transformer-based architecture of TARTE**. TARTE models a row in a table as a set of column($E$)–cell($X$) pairs. Given a tabular data with multiple data types, TARTE maps the representation of column and cell values to the same dimension using a language model ($LM$). From the mapped input, the transformer takes in a combination of both column and cell information to contextualize the cell content.

a domain defined by a topic, but tables from different sources. However, it uses a computationally expensive joint learning, where the model must be fine-tuned on the various datasets. To achieve full benefits of table foundation models, designing models that can be easily specialized to given domains or tasks is a priority.

# 3 TARTE: A backbone for knowledge pre-training

TARTE is an easily-reusable pre-trained model that encodes *data semantics* across heterogeneous tables by pre-training from large knowledge bases. This section details the main components of TARTE: (1) a transformer-based architecture that models the data semantics of table entries via the dependencies between columns and cells; (2) knowledge pre-training from rich background information stored in large knowledge bases; (3) effective post-training to reuse knowledge pre-training in diverse downstream tabular tasks.

## 3.1 A transformer-based architecture

As with the success in LLMs (Devlin et al., 2019; Achiam et al., 2023, ...) and pre-trained models for tabular data (*e.g.,* Hollmann et al., 2025; Kim et al., 2024), TARTE builds upon a variant of the transformer architecture (Vaswani et al., 2017). Designing such a variant entails *1)* a suitable transformation of input data to vector representations, for instance word tokenization and positional encoding in LLMs, *2)* a neural architecture based on the self-attention mechanism to capture the complex dependencies across inputs.

To learn across tables, a central challenge is to find a common representation of tables across heterogeneous datasets. Tables store diverse and incongruous information, despite the structured layout of rows and columns. To name a few challenges, tables represent information with different number of columns, data types (*e.g.,* numerical or discrete), and naming conventions (*e.g.,* France or FR). Combining column name with cell values enables representing diverse columns (see appendix A.1). Diverse data types call for tokenization of strings (Wang & Sun, 2022) or using a language model (Yan et al., 2024, discretizing numerical values). Additionally, Kim et al. (2024) represents each row in a table with a graph, where each cells and columns are represented with nodes and edges, respectively, with embeddings of strings (FastText, Mikolov et al., 2017).

TARTE borrows from Kim et al. (2024) the modeling of column($E$)–cell($X$) pairs but loosens the graph structure. Figure 1 shows the modeling process of a table entry to a suitable input for the transformer. Given a table of $k$ columns with multiple data types, the $i$-th row is a set $\{(E_j, X_j^i)\}_{j=1}^k$, in which all the components are mapped to the same dimension $d$ using a language model. More specifically, with column names $\{e_j\}_{j=1}^k$, a datetime converter, $D(\cdot)$, and a language model, $LM(\cdot)$, we have

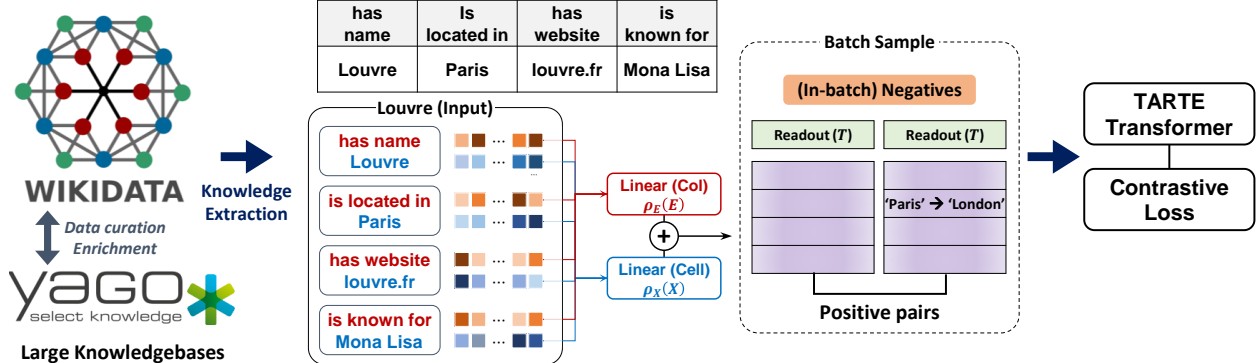

Figure 2: **TARTE Pre-training.** We extract facts from knowledge bases and replicate the input structure in Figure 1. Then, batches are constructed with positive samples by replacing parts of information (*e.g.*, Paris to London). TARTE is trained with contrastive learning using in-batch negatives (Chen et al., 2020b). The logos modified from YAGO and Wikidata, used under CC BY 4.0 and CC0, respectively.

$$E_j = LM(e_j) \quad \in \mathbb{R}^d,$$
$$X_j^i = x_j^i \cdot E_j \quad \in \mathbb{R}^d \quad \text{if } x_j^i \text{ is numerical,}$$

$$X_j^i = LM(x_j^i) \quad \in \mathbb{R}^d \quad \text{if } x_j^i \text{ is categorical/string,}$$
$$X_j^i = D(x_j^i) \cdot E_j \quad \in \mathbb{R}^d \quad \text{if } x_j^i \text{ is datetime.}$$

The language model enables TARTE to work with an open set of vocabulary, without requiring any intervention on string entries, hence bypassing the complex column or entity matching problems. It paves the way for TARTE to bring data semantics across tables (see subsection 4.3).

To obtain the transformer input $Z^i \in \mathbb{R}^{(k+1) \times d}$ for the $i$-th row, we simply add embeddings of linearly mapped column and cell information, $\rho_E(E)$ and $\rho_X(X)$, and stack to a learnable vector $T \in \mathbb{R}^d$:

$$Z^i = \texttt{stack}[T; \rho_E(E_j) + \rho_X(X_j^i)] \qquad \text{for column } j = 1, \dots, k$$

where $\rho(\cdot) = \texttt{Linear(ReLU(LayerNorm}(\cdot)))$ and $T \in \mathbb{R}^d$ works as the readout element (similar to the $\texttt{[CLS]}$ token in Devlin et al. (2019)). The processed input $Z$ is then fed to an encoder-based transformer with the typical multi-head self-attention and feed-forward module (Vaswani et al., 2017).

### 3.2 Knowledge pre-training from large knowledge bases

Figure 2 summarizes the overall pre-training process of TARTE. TARTE learns data semantics of heterogeneous tables by pre-training on large knowledge bases, containing millions of real-world facts.

**Pre-train data** Capturing tabular knowledge requires pre-training from diverse tabular data. While the best models for numerical data use synthetic data generation (Hollmann et al., 2023; 2025), Kim et al. (2024) have pushed the idea of pre-training from knowledge bases, due to their resemblance to tabular data. However, the high level of curation (particularly for YAGO3, Mahdisoltani et al., 2013) leads to datasets that lacks the diversity of relational (column) information found in real-world tables (see appendix A.3).

Going beyond Kim et al. (2024), TARTE expands the coverage of pre-train data by combining two large knowledge bases, YAGO4.5 (Suchanek et al., 2024) and Wikidata (Vrandečić & Krötzsch, 2014). In essence, YAGO4.5 is a cleaned version of Wikidata with a simplified taxonomy and much fewer properties to facilitate automated reasoning. To build our dataset, we restrict YAGO4.5 to entities with a Wikipedia page as these come with abundant information, valuable for pre-training. YAGO4.5 includes relatively little numerical information compared to downstream tables, and lacks diversity in relational information, especially for numerical triples. To address these shortcomings, we enrich with numerical facts (numbers and dates) from the more comprehensive but noisier Wikidata. The resulting knowledge base describes over 5.5 million entities and includes 30 million facts with 687 distinct relations (see appendix A.3).

**Preprocessing the pre-training data** The curated dataset is a set of triples with 'head', 'relation', and 'tail', $(h, r, t)$, in which $r$ and $t$ describe $h$. For example, a fact 'Louvre is located in Paris' is represented as $(h, r, t)$ where the elements are 'Louvre', 'is located in', and 'Paris', respectively (Figure 2). To enable pre-training, it requires additional steps of preprocessing for each data type of strings, numbers, and datetimes. For strings, we build a look-up table of embeddings built from a language model. In particular, we choose FastText (Mikolov et al., 2017): text entries in tables mostly contain one or a few words, too short for LLMs. For numerical values, including datetimes converted into fractional years, we perform relation-wise power transformation (Yeo & Johnson, 2000). Power transform has shown to be effective in numerous tabular learning studies, including pre-trained models (Hollmann et al., 2023; 2025; Kim et al., 2024).

**Batch sampling and contrastive loss** To construct a batch for TARTE pre-training, we first select $N_b$ entities and extract related facts for each. For example, if "Louvre" is selected, we extract its corresponding information, such as location, website, and its well-known art Mona Lisa (see Figure 2). In most cases, each entity would have different number of associated facts. However, there is only a fixed number of columns within a given table. Thus, we trim the number of related information for each entity to have a fixed number of facts across the batch. To enable contrastive learning, we also include positive samples, which are generated by replacing one or two of the facts with different information for each entity (see appendix A.4).

The embedding of a given table row is assembled with a linear readout $T$ from the output of the transformer. We then apply the contrastive learning framework of Chen et al. (2020b), in which other entities inside the batch are considered as negatives. In contrast to the widely-used cosine similarity, we take the Gaussian kernel with median distance as the bandwidth. We then use the InfoNCE contrastive loss (Oord et al., 2018).

**Model specifications** The transformer architecture of TARTE is specified as follows: We set three self-attention layers with 24 multi-head attentions, 768 hidden dimension, and 2 048 feed-forward dimension per layer. For the projection layers for contrastive learning, we set two linear layers with hidden and output dimensions of 2 048 and 768, respectively. The resulting model contains over 25 million trainable parameters.

### 3.3 Learning with the backbone: fine-tuned, or frozen, combined with another model

Given downstream tables[1], the knowledge backbone of TARTE facilitates learning through various post-training paradigms: fine-tuning, reused as a frozen featurizers, combined with other models.

**Fine-tuning a specific task** To fine-tune TARTE, we replace the projection layers for contrastive learning with three layers of $\rho(\cdot) = \texttt{Linear}(\texttt{ReLU}(\texttt{LayerNorm}(\cdot)))$, that focuses on task-specific settings. The model is trained end-to-end with parameters of the transformer layers kept frozen. Additionally, we adopt bagging (Breiman, 1996), which has been shown to be useful in several works for neural networks (*e.g.*, Kim et al., 2024; Holzmüller et al., 2024). For this, we train multiple models on different train-validation splits used for early-stopping, and average the outputs from each model to form predictions.

**TARTE as a table featurizer with frozen backbone** Similar to the sentence-transformers (Reimers & Gurevych, 2019) on LLMs, TARTE can be used to generate meaningful embeddings for table entries. The preprocessed input of a downstream table is passed through the frozen backbone of TARTE, and the embedded representation of the readout element $T$ (similar to the `[CLS]` token in LLMs) can be used with any machine learning model as a pipeline to make predictions on unseen data.

**Boosting a complementary model[2]** As TARTE is pre-trained from large knowledge bases, the embeddings of the TARTE featurizer potentially hold implicit background information. Such prior knowledge can be useful, especially in few-shot settings; but when the original table provides sufficient information for learning, the background information, on its own, becomes less useful for inference (see Figure 3). Yet, we argue that the implicit prior information continues to be useful when combined appropriately with a complementary tabular model. To accompany the prior to tabular models, we formulate a boosting strategy:

---

[1]Downstream tables are preprocessed as for pre-training (subsection 3.2): FastText string embedding, power transformation on numbers, datetime columns.

[2]We also experimented with various stacking approaches, but they did not bring the marked benefits of boosting.

the base tabular model with the original table is ensembled with a model that fits the (train) residuals of the base model with TARTE embedded features. For efficiency, we use TARTE with a Ridge regression.

**Specializing to a domain**   A paramount aspect of foundation models is to enable repurposing and specializing to a domain. For table foundation models, the requirement goes alongside with cross-table modeling van Breugel & van der Schaar (2024), to be able to draw from different tables. Fine-tuning across different tables within the same domain can work, but with a hefty cost (Kim et al., 2024). Rather, a specialized model that is easy to re-adjust would be ideal, *e.g.,* to compensate data drifts.

Here, the ability of TARTE as a table featurizer comes in handy. Avoiding a costly joint-learning (*e.g.,* Kim et al., 2024), TARTE can readily extract embeddings from fine-tuned models of related tables. The domain specialized representations can then be incorporated with the boosting strategy: the residuals are sequentially fitted with domain specialized representations. Boosting reuses the implicit information from tables within the same domain, embarking domain specialized predictions for downstream tables. For multiple source tables, the same process is repeated for each source table, in a multi-step boosting.

### 3.4   Differences to the CARTE approach

While TARTE draws from CARTE (Kim et al., 2024), we will see that its representations capture much more knowledge from pre-training, providing value without fine-tuning. Multiple differences improve pre-training.

First, TARTE avoids the graph structure used in CARTE. This graph structure, used as a data representation across tables, creates a bottleneck that limits the flow of information between inputs in the self-attention layers (Alon & Yahav, 2021). For example, representing the entry in Figure 1 with the graph structure of Kim et al. (2024) masks the attention between 'Country' U.S. and 'Language' English. Indeed, in CARTE, these inputs are connected only via the center node, a row-summary token. On contrary, TARTE keeps all such relations in a single attention mechanism, providing context to capture data semantics.

Second, TARTE is trained as a rather shallow architecture of three layers, reflecting that the input for the transformer works in a column-level setting. For both CARTE and TARTE, the columns work similarly as the context-window for LLMs, in which the row-dimension of the input is determined by the number of columns. In many cases, the number of columns in tables are relatively limited, and thus the models are more prone to the problem of oversmoothing (Chen et al., 2020a; Nguyen et al., 2023) with deeper architectures.

Third, TARTE reflects better downstream tasks by handling different data types and better pre-training data sources (see subsection 3.2 and appendix B.1), carefully preprocessing the batch samples, such as trimming to match the number of columns and limiting redundant relations. In addition, TARTE carefully controls the training procedures (see appendix A.4).

## 4   Empirical study: TARTE improves prediction and speed, and is reusable

### 4.1   Experimental set up: tabular learning

**Datasets and methods**   We use the benchmark from Kim et al. (2024): 40 regression and 11 classification datasets[3]. These tables come with informative columns and discrete entries. For comparing methods, evaluate post-training paradigms of TARTE (subsection 3.3) and the best performers in related benchmarks: the leading gradient-boosted tree models, XGBoost (**XGB**, Chen & Guestrin, 2016) and **CatBoost** (Prokhorenkova et al., 2018); neural-network model, **RealMLP** (Holzmüller et al., 2024); pre-trained models, **TabPFNv2** (Hollmann et al., 2025) and **CARTE** (Kim et al., 2024) with fine-tuning and boosting (**CARTE–B–**); simple linear model **Ridge** from scikit-learn (Pedregosa et al., 2011).

In terms of data preparation, most models provide native handling of diverse data types. However, for models that explicitly require numerical tables, we rely on heuristics provided by the `TableVectorizer` (**TabVec**) functionality of the skrub software. While TabPFNv2 handles readily the input tables, we optionally combine it with the `TableVectorizer`, for better handling of high-cardinality string and datetime columns.

---

[3]Datasets available at https://huggingface.co/datasets/inria-soda/carte-benchmark

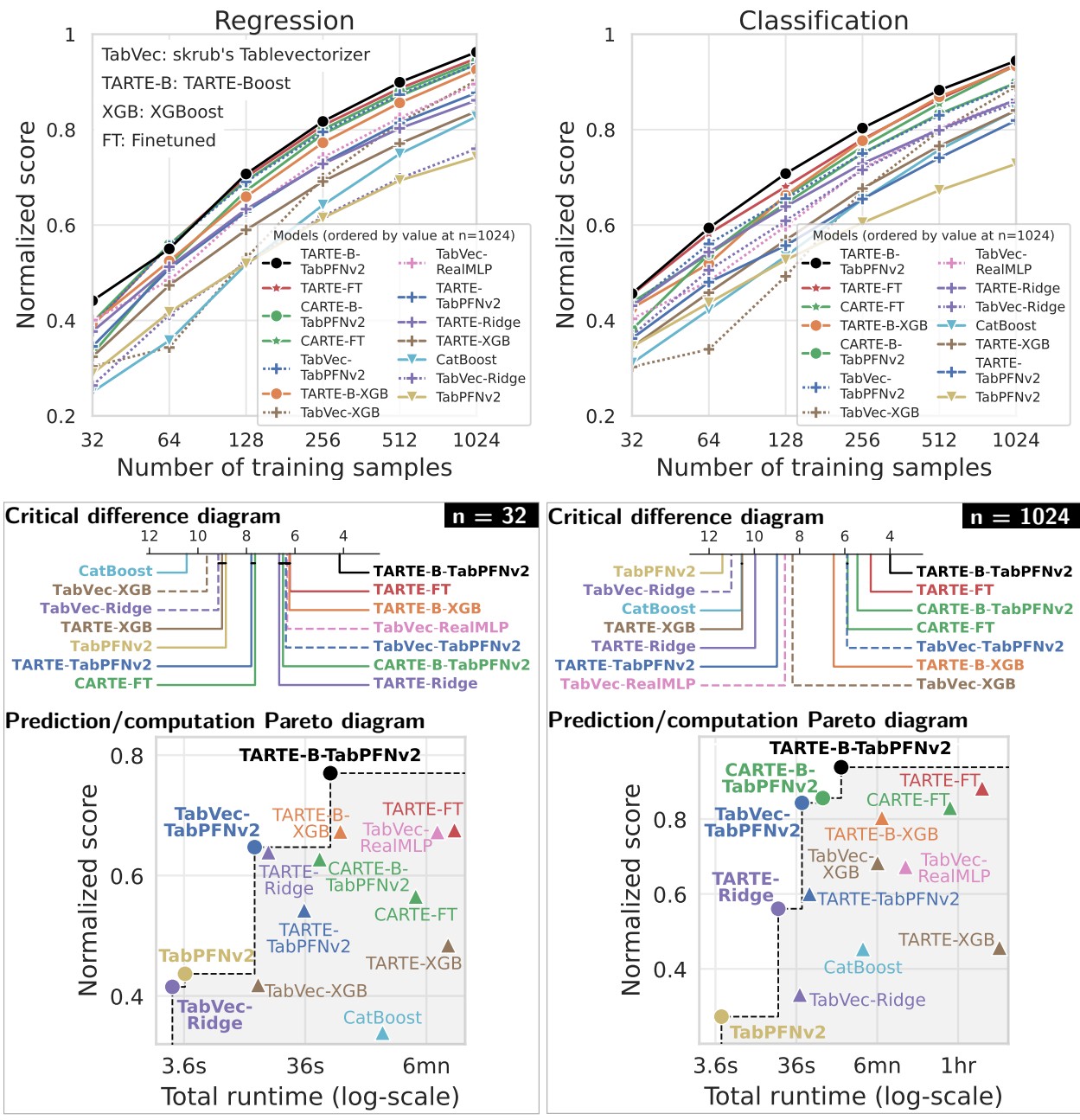

Figure 3: **TARTE performs best for learning on small tables** – **Top: Learning curve** for normalized prediction scores for regression and classification. In general, pre-trained models perform better, with variants of TARTE surpassing all baseline models. **Middle: Critical difference diagram of average rank** at $n = 32$ and $1\,024$. **Bottom: Pareto diagrams** normalized prediction scores with respect to total runtime (log-scale). Efficient base models bring runtime benefits, and TARTE brings additional performance gains.

For TARTE, we abbreviate the variants as follows:

- **TARTE − FT** : Fine-tuning TARTE on downstream tables.
- **TARTE −** : TARTE as table featurizer. We consider Ridge, XGB, and TabPFNv2 as prediction models.
- **TARTE − B −** : Boosting scheme with TARTE embedded features. We consider state-of-the-art tabular models, TabPFNv2 and XGB, as base models where `TableVectorizer` is used for data preparation.

Specific details on experiment settings (*e.g.,* hyperparameter selection) are presented in appendix B.1.

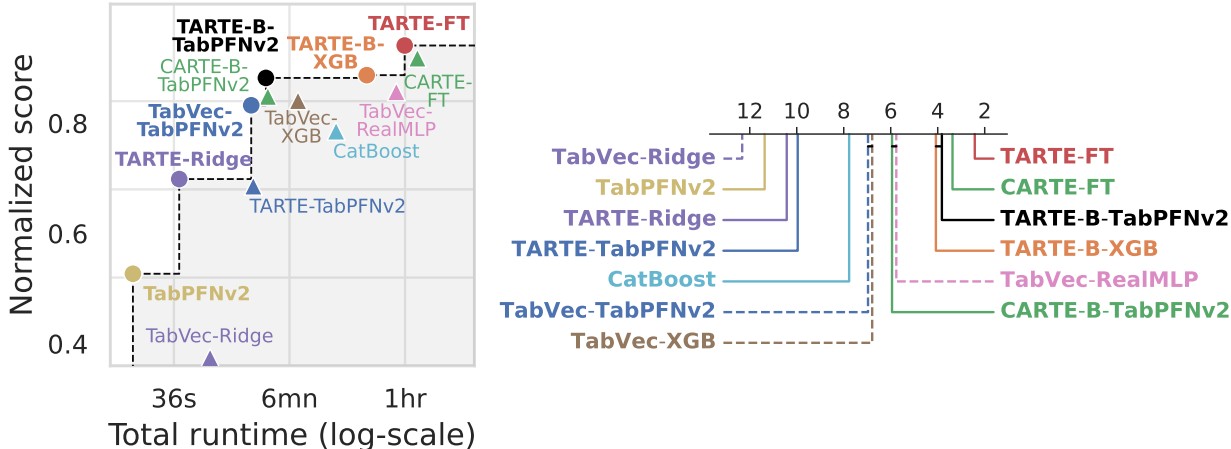

Figure 4: **Results for** $n = 10\,000$ – **Left: Pareto diagram** – **Right: critical difference diagram of average rank** Fine-tuned TARTE and TARTE boosting continues to surpass the baselines, but fine-tuning TARTE brings more benefits than on smaller data.

## 4.2 TARTE boosts learning, and can be computationally efficient

**On small tables: few-shot learning** Figure 3 (top) shows learning curves of normalized prediction scores as a function of sample sizes. The scores are normalized across all train sizes per dataset, with 1 as the best and 0 as the worst performing model. Pre-trained models (TARTE, CARTE, and TabPFNv2) generally perform better, with variants of TARTE as the best performing models, regardless of the sample sizes. Critical difference diagrams[4] (Figure 3, middle) show that the benefits brought by TARTE are significant.

Considering computation-time trade-offs, Pareto diagrams (Figure 3, bottom) show that variants of TARTE and TabPFNv2 form the Pareto frontier. When compute time is important, models that avoid fine-tuning (as TARTE–B–TabPFNv2 or TabVec–TabPFNv2) gain advantage, especially with larger $n$ (*e.g.*, $n = 1\,024$).

In general, boosting a base model with pre-trained embeddings (TARTE–B and CARTE–B) improves prediction. However, as TARTE is better pre-trained (see subsection 4.3), its embeddings complement better the base models, bringing prior information acquired from knowledge pre-training. The best performances require a strong base tabular model, as with TabPFNv2 and XGB. On the other hand, TARTE–FT does not have such requirements, but it comes with computation burdens, with extensive hyperparameter tuning[5]. Finally, TARTE featurizer is a good table preparator: compared to TabVec (dashed lines), it can markedly increase performance, bringing, for instance, ridge to a good position.

**On larger tables, TARTE helps both for prediction and scalability** Figure 4 summarizes prediction accuracy and runtime costs at $n = 10\,000$. Similar to small tables, pre-training schemes are the Pareto frontier, with TARTE–FT and TARTE–B outperforming baselines. TARTE–B continues to blend well with base models, TabPFNv2 and XGB. Fine-tuning TARTE brings more prediction benefits, but at a heavy cost with a 14-fold runtime increase compared to TARTE–B–TabPFNv2. TARTE–B–TabPFNv2 gives both performance and scalability: compared to TabVec–TabPFNv2, the prior from knowledge pre-training improves performance at a small compute cost (1 mn).

**On more numerical tables, comparing to TabLLM** We evaluate TARTE–FT and TARTE–B over nine datasets presented in TabLLM (Hegselmann et al., 2023). The datasets contain larger fraction of numerical columns with lower cardinality in categorical columns (Kim et al., 2024). Figure 5 shows methods comparison, as a critical difference diagram. In addition to four baselines from Hegselmann et al. (2023)

---

[4]Critical difference diagrams display average rank across models with crossbars depicting the two having no statistically significant difference based on Conover post hoc test after a Friedman test for pairwise significance (Conover, 1999).

[5]Due to the computation costs, the search space for TARTE-FT is limited (see Table 3)

Figure 5: **Comparison of baselines on more numerical tables**. For datasets with higher fraction of numerical columns (from Hegselmann et al. (2023)), TARTE–B continues to help base models for prediction.

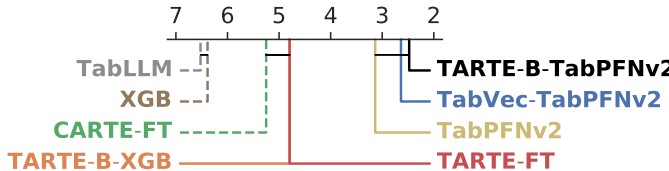

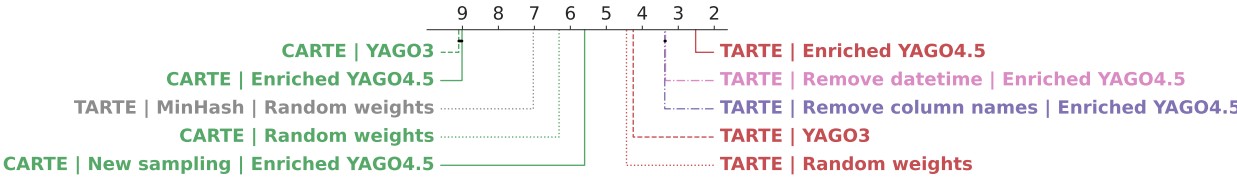

Figure 6: **Ablating architecture, pre-training, and preprocessing components**. A ridge is fitted with embeddings from different schemes: random weights (no pre-training); replacement of FastText with skrub's "MinHash" encoder; TARTE and CARTE pre-trained with YAGO3 and Enriched YAGO4.5; CARTE with TARTE pre-training schemes (New sampling); TARTE without datetime detection or column information.

and Kim et al. (2024) (in dashed lines), we include two TabPFNv2 variants considered in this study. While TabPFN tends to performs better, TARTE–B can help the base models, even on more numerical tables.

### 4.3 Good knowledge pre-training, adapted to downstream tasks, is important

**Better knowledge pre-training, better performance**  TARTE embeddedings boost prediction. But what drives this boost? Is it the inductive bias of the architecture, or is the pre-training on the knowledge base actually important? To answer this question, we investigate TARTE pre-training on small tables ($n = 32$ to $1\,024$), where the effect of TARTE is most eminent. Figure 6 presents the critical difference diagram for Ridge fitted with embeddings from different schemes. Here, CARTE with "New sampling" corresponds to CARTE architecture with TARTE pre-training schemes, and "MinHash" replaces FastText by MinHash encoding, that only captures morphological similarities of strings (Cerda & Varoquaux, 2020).

First, comparing MinHash to TARTE reveals the importance of FastText. A simple use of FastText with random (non pre-trained) weights already exhibit relatively strong performances. As FastText captures semantic similarities, its combination with a suitable transformer architecture (subsection 3.1 or CARTE) forms an inductive bias of *smoothness*: similar tables have similar representations.

Going further, a proper combination of pre-train data and procedures, the architecture, and preprocessing of tables improves knowledge pre-training (TARTE with Enriched YAGO4.5). For CARTE representations used without fine-tuning, pre-trained representations perform less well compared to random weights, regardless of the pre-train data: these representations lack out-of-the-box re-usability. TARTE pre-training schemes on CARTE (New sampling) helps but the performance does not match that of TARTE, highlighting the various sources of improvements through TARTE (see subsection 3.2). In addition, without datetime detection or column information, properly pre-trained weights still provide competitive representations. This suggests that TARTE can perform well even on tables without meaningful column names.

**Which downstream tables benefit from TARTE?**  Tables are very heterogeneous, and not all may correspond to learning tasks that benefit from the knowledge embarked in TARTE. We explore which characteristics of a downstream table leads to a performance boost with TARTE. Table 1 provides a multivariate analysis of various factors (dataset meta-features) that affect the prediction on downstream tables (linear model explaining the performance boost of TARTE on top of TabVec-Ridge).

We find that TARTE works less well on tables with long string entries (likely a limitation of FastText embeddings, as discussed in appendix A.2). However, it brings more benefits for tables with many strings considered as inliers to the pre-training source (using a One-Class SVM fitted on the embeddings of the pre-training strings to define an inlier score). TARTE also works better with fewer low-cardinality columns

Table 1: **Factors of success for TARTE.** Coefficients and their confidence intervals of a linear model explaining the improvement of TARTE–Ridge over TabVec–Ridge across datasets (R-squared: 0.22). Negative values mean that greater values for the corresponding meta-feature are associated to less benefits of TARTE. Performance decrease for long strings suggest a limitation of the language model, FasText. Importance of inlier probability shows the need for pretraining to cover well downstream terminology.

| Dataset feature | Coef. | **CI**[0.025, 0.975] |
|---|---|---|
| Avg. string length | -0.35 | [-0.40, -0.31] |
| Avg. inlier prob. | 0.29 | [0.26, 0.33] |
| # datetime cols | 0.28 | [0.24, 0.31] |
| Avg. string sim. | 0.18 | [0.13, 0.22] |
| # low card. cols | -0.18 | [-0.23, -0.14] |
| # numerical cols | 0.13 | [0.09, 0.16] |
| # high card. cols | 0.06 | [0.02, 0.10] |
| log(train-size) | 0.04 | [0.00, 0.08] |

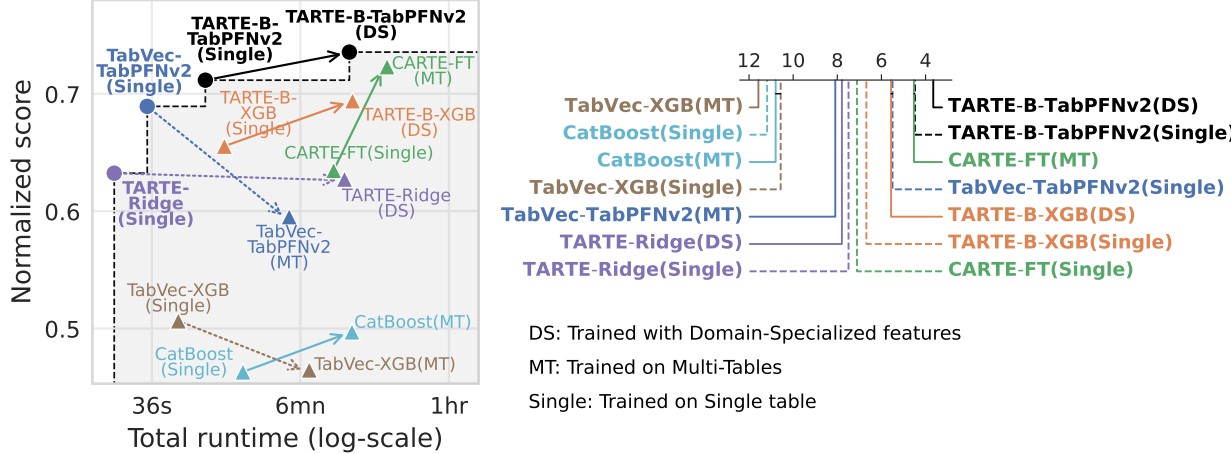

Figure 7: **Domain specialization from a single source − Left: Pareto diagram − Right: critical difference diagram of average rank.** 'DS' and 'MT' denote Domain-Specialized and Multi-Tables schemes, respectively. Models that blends inference from the target table with representations tuned on the source improve (TARTE–B and CARTE). TARTE–B–TabPFNv2 gives the best predictions.

(trees shine on these), with many datetime columns, with strings similar across rows, with more numerical or high-cardinality columns, and with more data. Overall, we find that TARTE brings benefits on more complex tables, with many complex strings, but particularly so if these look like strings seen during pre-training.

### 4.4 Specializing to a domain: fine-tuned TARTE gives representations that transfer

**Experimental set up and baselines** We investigate whether reusing TARTE representations *from fine-tuned models* facilitates subsequent learning. We evaluate TARTE–Ridge and TARTE–B with TabPFNv2 and XGB. Here, TARTE–Ridge denotes a Ridge model fitted on fine-tuned TARTE embeddings and TARTE–B sequentially fits the residuals with domain specialized representations (see subsection 3.3). As baselines, we consider top-performing tabular models that provide native handling of missing values. Indeed, even with differing columns across tables, these models can be fitted with a stacked table. We also include CARTE multi-table (CARTE-MT), the only baseline, to our knowledge, that learns across tables without correspondences (Kim et al., 2024). Note that the setup for TARTE and baseline models is slightly different. Baselines fit all tables, sources and target, jointly; while the TARTE model does not need access to the source tables to be reused for transfer. We consider each domain one after the other, and for each, we consider every combination of source tables and a target table: source tables to specialize the model, and target table to evaluate the corresponding learner. As datasets, we select from the previous set of tables group of tables within the same domain, but acquired in different settings (Kim et al., 2024). See appendix B.3 for details.

**TARTE improves with domain specialized representations** Figure 7 presents the improvements and runtime costs for a single source table. Here, the abbreviations 'DS' and 'MT' denote Domain-Specialized and

Figure 8: **Domain specialization from multiple sources: comparison between TARTE and CARTE.** Improvements from the best performing model without any source information, TARTE–B–TabPFNv2, with respect to the total runtime across different train sizes. While both TARTE and CARTE improve with multiple sources, TARTE is far more efficient. In contrast to CARTE, TARTE can reuse the fitting on the source tables (here, 20 mn on average).

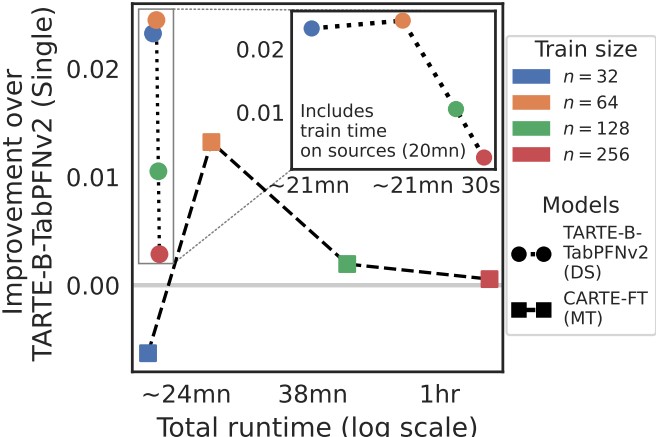

Multi-Tables, respectively. Out of top-performing tabular models, those that actually benefit are TARTE–B, CARTE, and CatBoost. Here, most of the runtime of TARTE is driven by the training time of the source model. The cost can be amortized if we are given a new target table: TARTE can readily reuse the domain-specialized models, which gives an advantage over baselines that requires a new fitting of the model.

**TARTE stays efficient with multiple sources**   We evaluate every possible source–target combinations with TARTE–B–TabPFNv2(DS) and CARTE-MT. Figure 8 gives the improvements of respective models compared to the best performing model on single tables (TARTE–B–TabPFNv2), with respect to the runtime. Both models benefit from source tables, with larger gains for TARTE–B–TabPFNv2(DS). Compared to Kim et al. (2024), the baseline here is much more performant, and thus it is more difficult to improve. Concerning the runtime, CARTE-MT requires pairwise fitting for each source, markedly increasing compute costs even for small increase in train-size. TARTE, however, is more efficient: the runtime of TARTE is dominated by the training time of source tables (20 mn). Therefore, reusing domain-specialized models imposes far less compute cost, putting TARTE in a better position for transfer within the specific domain.

## 5   Discussion and conclusion: A knowledge backbone that can be reused

Foundation models have changed the landscape of machine learning because they facilitate a huge variety of applications and can efficiently lead to specialized models. Table Foundation Models are progressing rapidly, one line of work improving table-level tasks such as table understanding and question answering (Zhang et al., 2024; Li et al., 2024), another geared toward tabular learning, *i.e.,* row-level predictive analytics (Hollmann et al., 2025; Kim et al., 2024). In the latter, the downstream task comes with strong supervision. Here, developing foundation models requires extra work to fulfill the promises of pre-training and re-use.

**Knowledge pre-training that helps tabular learning**   Our study shows the value of pre-training to acquire world knowledge for tabular learning (Figure 6). Current table understanding models clearly leverage world knowledge, but it is less visible in tabular learning ones. Table understanding models are typically variants of LLMs, generative, pre-trained an large textual corpora, and often used via prompting and thus in context learning. For tabular learning, however, post-training for the downstream tasks is typically further away from the next-token prediction used to inject world knowledge in LLMs. Indeed, for tables with many rows, a good tabular learner often needs to aggregate information across these rows, as opposed to retrieval of a few rows which suffices for typical table-understanding tasks. But language models do not leverage long contexts uniformly (Liu et al., 2024). By matching pre-training tasks to downstream tasks, the TabPFN literature has managed to train suitable in-context models, valuable for tabular learning. However, this has required synthetic datasets that do not bring world knowledge.

Powerful post-training used for tabular learning can compensate for suboptimal pre-training. In fact, models without pre-training (such as XGBoost) are very strong baselines. Also, our results (Figure 6) show that CARTE, a pretrained model for tabular learning, achieved its good performance thanks to post-training,

namely fine-tuning. A drawback of requiring sophisticated post-training is the compute cost. Investing in pre-training, as with TARTE, enables to use simple downstream learners, as a Ridge, which becomes a very strong baseline with TARTE (On small, but also mid-sized data: figures 3 and 4). Pre-training cost is then "amortized" by making downstream learning and inference cheap.

**A re-usable backbone improves predictions and decreases costs**   TARTE's backbone is suited to many post-training approaches. Well pre-trained, its transformer-based architecture gives a good encoder for complex tables that can be easily re-used and specialized as a backbone. It can be plugged into any learner, possibly cached to facilitate operations. Fine-tuning to a domain gives best prediction. But different choices of post-training (using TARTE only as data preparation for a subsequent learning, boosting, or fine-tuning) explore different trade-offs in prediction performance versus computational costs. Using TARTE improves the "Pareto optimality": more prediction performance for the same computational cost. This approach is scalable and beneficial even for larger tables. Finally, while strings (in column names and cell entries) are central to TARTE, the encoder also leads to state-of-the-art performance on more numerical tables.

A popular aspect of foundation models is their ability to be specialized, as visible from the number of fine-tuned models on hugging face (*e.g.,* 1 566 derived from Roberta, as of May 2025, huggingface). A fine-tuned TARTE can be readily reused for multiple applications in the given domain, improving downstream prediction and computational performance. Such fine-tuning followed by an independent reuse is to be contrasted with most prior successes of transfer in tabular learning, built on joint learning and thus increasing operational costs rather than decreasing them. On the contrary, reusing a domain-specialized version of TARTE uses *less* computational resources than starting from scratch or fitting a cross-table model.

The agenda of table foundation models calls for repurposable and reusable models that are as powerful and easy to apply to new tables as possible. Our study makes a step is this direction, showing how to pretrain to capture world knowledge and reuse it for tabular learning.

## Acknowledgements

The authors acknowledge the support in part by the French Agence Nationale de la Recherche under Grant ANR-20-CHIA-0026 (LearnI). We also would like to thank the anonymous reviewers for constructive comments and valuable suggestions, which have significantly improved the quality of the manuscript.

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

Table 2: Comparison of pre-train datasets for TARTE and CARTE.

| Data | Used in | # Entities | # Relations | # Facts |
|---|---|---|---|---|
| YAGO3 | CARTE | 4 027 996 | Total: 65
Categorical: 38
Numerical: 27 | Total: 18 108 790
Categorical: 14 826 655
Numerical: 3 282 135 |
| YAGO4.5 | - | 5 415 689 | Total: 97
Categorical: 72
Numerical: 25 | Total: 23 457 263
Categorical: 16 296 953
Numerical: 7 160 310 |
| YAGO4.5 Enriched | TARTE | 5 576 475 | Total: 687
Categorical: 72
Numerical: 615 | Total: 30 262 472
Categorical: 16 296 953
Numerical: 13 965 519 |

# A  Further details on backbone and pre-training

## A.1  Modeling with column−cell pairs: context-aware for transformers

Given a table, TARTE models with a set of column−cell pairs and combine the embeddings to form the input of the transformer. The column information is crucial to supplement context for the transformers (Kim et al., 2024). For instance, the entry (*e.g.,* 'A.I') or the content of the table (*e.g.,* 'movies') would be difficult to understand without the column information (see Figure 1). For knowledge bases of the pretrain data, this corresponds to modeling relations, which has been pivotal to knowledge embedding models (Cvetkov-Iliev et al., 2023). Moreover, the column embeddings serve as a medium to combine different data types for a uniform processing of entries.

## A.2  Language model used in TARTE

We use FastText embeddings (Mikolov et al., 2017), which hold some limitations for long string entries (subsection 4.3). Yet, string (text) entries in tables mostly contain only a few words: across 51 datasets, the median ratio of unique entries with more than 10 words is 0.025. An interesting alley to explore more sophisticated language models, possibly using different language models depending on different structure of strings (e.g., simple words or sentences).

## A.3  Pre-train data

Table 2 gives the statistics of the pre-trained datasets used by TARTE and CARTE. They both describe approximately the same number of entities–those that have a Wikipedia page (Suchanek et al., 2024). However, the pre-train dataset for TARTE is enriched and diversified with more numerical information: YAGO4.5 Enriched, used in TARTE, describes almost twice as many numerical facts and, most importantly, includes far more diverse properties (615 distinct relation types against 27 for YAGO3).

## A.4  Pre-training procedures

**Training specifications**  The batch size is set as 512 in which 256 entities are randomly selected with one additional positive for each. The total number of steps for training is 200 000, with the AdamW optimizer and the cosine scheduler. The learning rates were set as $lr_{min} = 10^{-8}$, $lr_{max} = 10^{-6}$ of warm-up over the first 2 000 steps, followed by a linear decay in learning rate schedule. The probability for all dropout layers was set as 0.1.

**Batch sampling**  To enable contrastive learning, we construct batch samples by generating positive samples that replace parts of information (for instance, replacing Paris with London, see Figure 2). While this may be counterintuitive, it helps the pre-train model to embed representations that is geared for capturing

smoothness (similar rows having similar target value) in a table. When trimming relations to have a fixed number of related information across entities for a batch, we consider the number of unique relations for each entity. For example, a well-known city can have many number of relations 'has neighbor', which dominate the facts about the city. This creates the difficulty in resembling the input data with a table, and thus we trim the relations such that there are not many duplicate relations for an entity.

**Contrastive loss with Matryoshka embeddings**   For the contrastive learning, we use the Matryoshka embedding scheme (Kusupati et al., 2022) to provide various pretrained embededdings of reduced dimensions. The Matryoshka embeddings attach several linear projection layers of different dimensions in parallel, calculate contrastive loss for each dimension, and aggregate the losses to backpropagate through the whole network. For TARTE pre-training, the dimension set was {64, 128, 256, 512, 768}.

## B   Details on downstream experiments

### B.1   Training procedures

**Data preparation**   We use TARTE embeddings of dimension, $d = 768$ except for TabPFNv2, which we set $d = 256$ since the maximum feature-size that TabPFNv2 can handle is 500. For `TableVectorizer` from the skrub, categorical columns are differently encoded depending on the cardinality (number of categories): Columns with low cardinality are one-hot encoded while those with high cardinality are encoded using the Gamma-Poisson encoder (Cerda & Varoquaux, 2020). For models without native handling missing values, we imputed with the mean for numerical features, and treated as another category for categorical features.

**Hyperparameter optimization**   We perform hyperparameter selection, except for TabPFNv2. For fine-tuned models, TARTE–FT and CARTE–FT, we use a custom grid-search hyperparameter optimization function tailored with the bagging from different train/validation splits. It identifies the optimal hyperparameters based on mean validation loss. For Ridge regression, we use `RidgeCV` from scikit-learn (Pedregosa et al., 2011), which performs an efficient Leave-One-Out Cross-Validation to select the hyperparameters. For rest of the models, we run 5-fold cross-validation over 100 random search iterations. Table 3 shows the hyperparameter spaces for each method. Most of the spaces were adapted from Grinsztajn et al. (2022) and Holzmüller et al. (2024), except for the fine-tuned models, which were adapted from Kim et al. (2024).

**Additional details**   For evaluation on single tables, the train-size for each tables varied from 32, 64, 128, 256, 512, 1 024, and 10 000; remaining data was set as the test set. Out of 51 datasets, 27 datasets contain sufficient data points for the train-size of 10 000. The performance was measured with $R^2$ score for regression and the Area Under Receiver Operating Curve (AUROC) for classification tasks. For the runtime, it measures the total time for data preparation, hyperparameter optimization, and prediction. Overall, the results were recorded on 10 different train/test splits for each dataset.

### B.2   Details on multivariate analysis

The inlier score is obtained by fitting a One-Class SVM on the embeddings of strings present in the pre-train data. Considering the size of the pre-train data, we train `SGDOneClassSVM` from scikit-learn (Pedregosa et al., 2011), that solves linear One-Class SVM using stochastic gradient descent. In regards to the categorical columns, we follow the heuristics from `TableVectorizer` in skrub package, with the criterion of high cardinality set as 40.

### B.3   Experimental set-up for domain specialization

We consider the few-shot settings with the train-size on the target table varied from 32, 64, 128, and 256. The splits were set as same as that of singletables to enable comparable results. For each baseline, some additional details were considered.

Table 3: Hyperparameter space for models considered in the study.

| Methods | Parameters | Grid |
|---|---|---|
| TARTE–FT | Learning rate | $[1, 2.5, 5, 7.5, 10] \times 1e^{-4}$ |
| | Batch size | 16 for $n < 10\,000$ else 256 |
| CARTE–FT | Learning rate | $[1, 2.5, 5, 7.5, 10] \times 1e^{-4}$ |
| | Batch size | 16 for $n < 10\,000$ else 256 |
| Ridge | Alpha | $[1e^{-2}, 1e^{-1}, 1, 1e^1, 1e^2]$ |
| CatBoost | Num. estimators | 1000 |
| | Max depth | UniformInt [2, 6] |
| | Learning rate | LogUniform $[1e^{-5}, 1]$ |
| | Bagging temperature | Uniform [0, 1] |
| | $l_2$-leaf regularization | LogUniform [1, 10] |
| | One hot max size | UniformInt [0, 25] |
| | Random strength | UniformInt [1, 20] |
| | Leaf estimation iterations | UniformInt [1, 20] |
| | od_wait | 300 |
| | od_type | 'Iter' |
| XGBoost | Num. estimators | 1000 |
| | Max depth | UniformInt [2, 10] |
| | Learning rate | LogUniform $[1e^{-5}, 1]$ |
| | Min child weight | LogUniform [1, 100] |
| | Subsample | Uniform [0.5, 1] |
| | Colsample by level | Uniform [0.5, 1] |
| | Colsample by tree | Uniform [0.5, 1] |
| | Gamma | LogUniform $[1e^{-8}, 7]$ |
| | Lambda | LogUniform [1, 4] |
| | Alpha | LogUniform $[1e^{-8}, 100]$ |
| | Early stopping rounds | 300 |
| RandomForest | Num estimators | UniformInt [50, 250] |
| | Max depth | [None, 2, 3, 4] |
| | Max features | [sqrt, log2, None, 0.1, 0.2, 0.3, 0.4, 0.5, 0.6, 0.7, 0.8, 0.9] |
| | Min samples leaf | LogUniform [1.5, 50.5] |
| | Bootstrap | [True, False] |
| | Min impurity decrease | [0, 0.01, 0.02, 0.05] |
| MLP | Num layers | UniformInt [1, 4] |
| | Layer size | UniformInt [16, 1024] |
| | Dropout | Uniform [0, 0.5] |
| | Learning rate | LogUniform $[1e^{-5}, 1e^{-2}]$ |
| | Weight decay | LogUniform $[1e^{-8}, 1e^{-2}]$ |
| | Batch size | [16, 32] |

- **TARTE**: The runtime of TARTE models include the training time of source tables. For each source table, we fine-tuned the pre-trained TARTE with the associated task. No hyperparameter tuning was performed, with the batch size and the learning rate set as 256 and $5e^{-4}$, respectively.
- **CARTE-MT**: We follow Kim et al. (2024) for the set-up. The runtime includes training without source (including hyperparameter optimization) and joint learning of target-source pairs (Kim et al., 2024).
- **Other baselines**: For models that stack the target and a source table, the maximum size of the training data (including both the target and the source table) was set as $10\,000$, which is the boundary for TabPFNv2, and larger size would incur overfitting to the source (Kim et al., 2024). Moreover, the column names are matched through manual inspection for best possible performances.

**Datasets** For datasets from a similar domain, we adapt those used in multi-table learning in Kim et al. (2024). These are tables within the same domain, but acquired in different settings (for instance from multiple sources). We have 12 domains: Wine prices (4 tables), Wine ratings (3 tables), Beers (2 tables), Used cars (5 tables), Films (2 tables), Dramas (2 tables), Anime (2 tables), Baby products (2 tables), Bike sales (2 tables), Employee remunerations (3 tables), Restaurant ratings (3 tables), Journal scores (2 tables).

## C    Hardware specifications

The hardware specifications for pretraining and downstream tasks are as follows. For the pre-training of TARTE, a single NVIDIA A40 (48GB) gpu was used. The downstream experiments was run on 32 cores of CPU except for TabPFNv2 variants for all $n$, and TARTE–FT and CARTE–FT at $n = 10\,000$, in which we used gpus. The hardware was chosen based on availability.

- **GPUs**: NVIDIA V100 (32GB VRAM), A40 (40GB / 48GB VRAM)
- **CPUs**: AMD EPYC 7742 64-Core Processor, AMD EPYC 7702 64-Core Processor (512GB RAM), Intel(R) Xeon(R) CPU E5-2660 v2, Intel(R) Xeon(R) Gold 6226R CPU (256GB RAM)

## D    Implementation of TARTE

The implementation of TARTE is available at `https://github.com/soda-inria/tarte-ai`.

## E    Extended results

### E.1    Results on small tables

Figure 9 presents the results for all train-sizes in few-shot setting. We include additional baselines of Random Forest (**RF**), the classical Multilayer Perceptron (**MLP**), and TARTE–CatBoost using the default hyperparameters. TARTE, through various post-training schemes, outperforms the baselines regardless of train-sizes. In addition, TARTE featurizer outperforms TableVectorizer (except for TabPFNv2) for a given learner in small training samples ($n \leq 256$). For TARTE–TabPFNv2, it underperforms likely due to that the dimension of TARTE embeddings was 256 for TabPFNv2 and the resulting data representations do not have a distribution matching to that of synthetic tables from TabPFNv2 pre-training.

### E.2    Results on single source domain specialization

Figure 9 shows improvements for using a single source table. Regardless of the train-sizes, TARTE–B provides benefit to the base models. Compared a CARTE–MT with the joint learning scheme, TARTE shows the benefit of having a reusable model. Once specialized models is available, TARTE can readily benefit without requiring complex refitting of the model, placing TARTE in a better position for transfer.

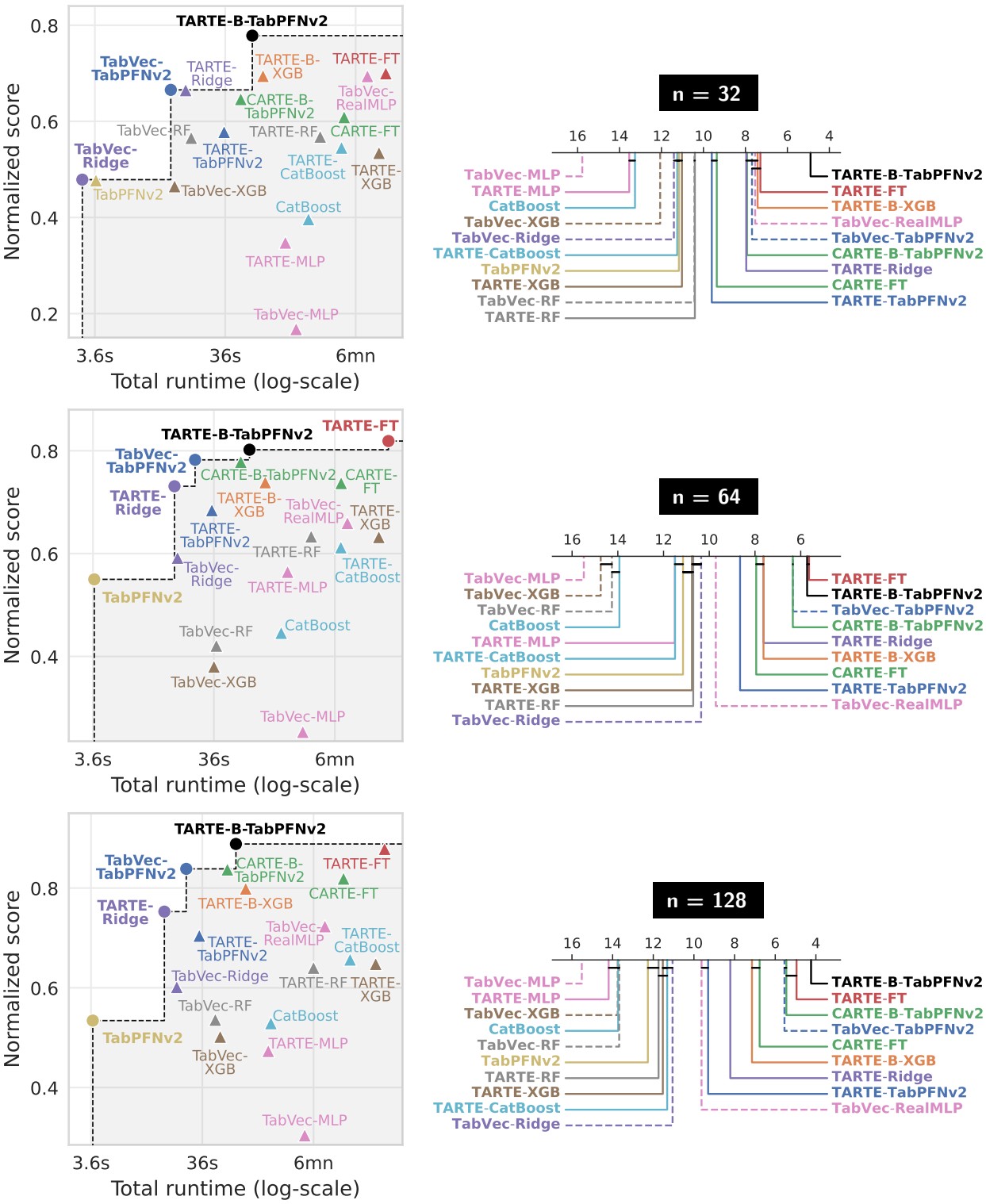

Figure 9: **Results for small tables** – **Left: Pareto diagram** – **Right: critical difference diagram of average rank** TARTE surpasses the baselines, and can act as an effective table preparator, especially for small number of training samples ($n \leq 256$).

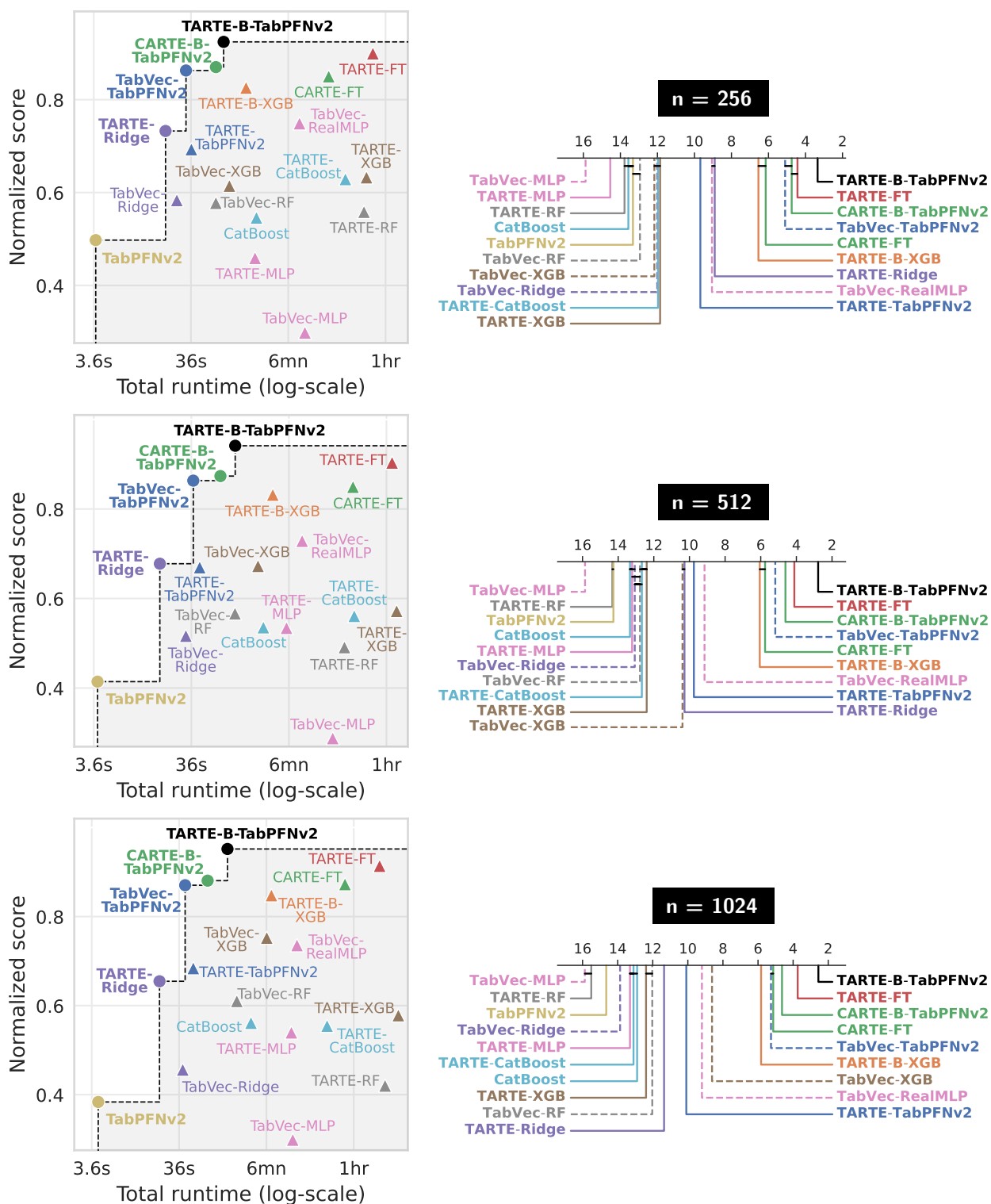

**Continued.**

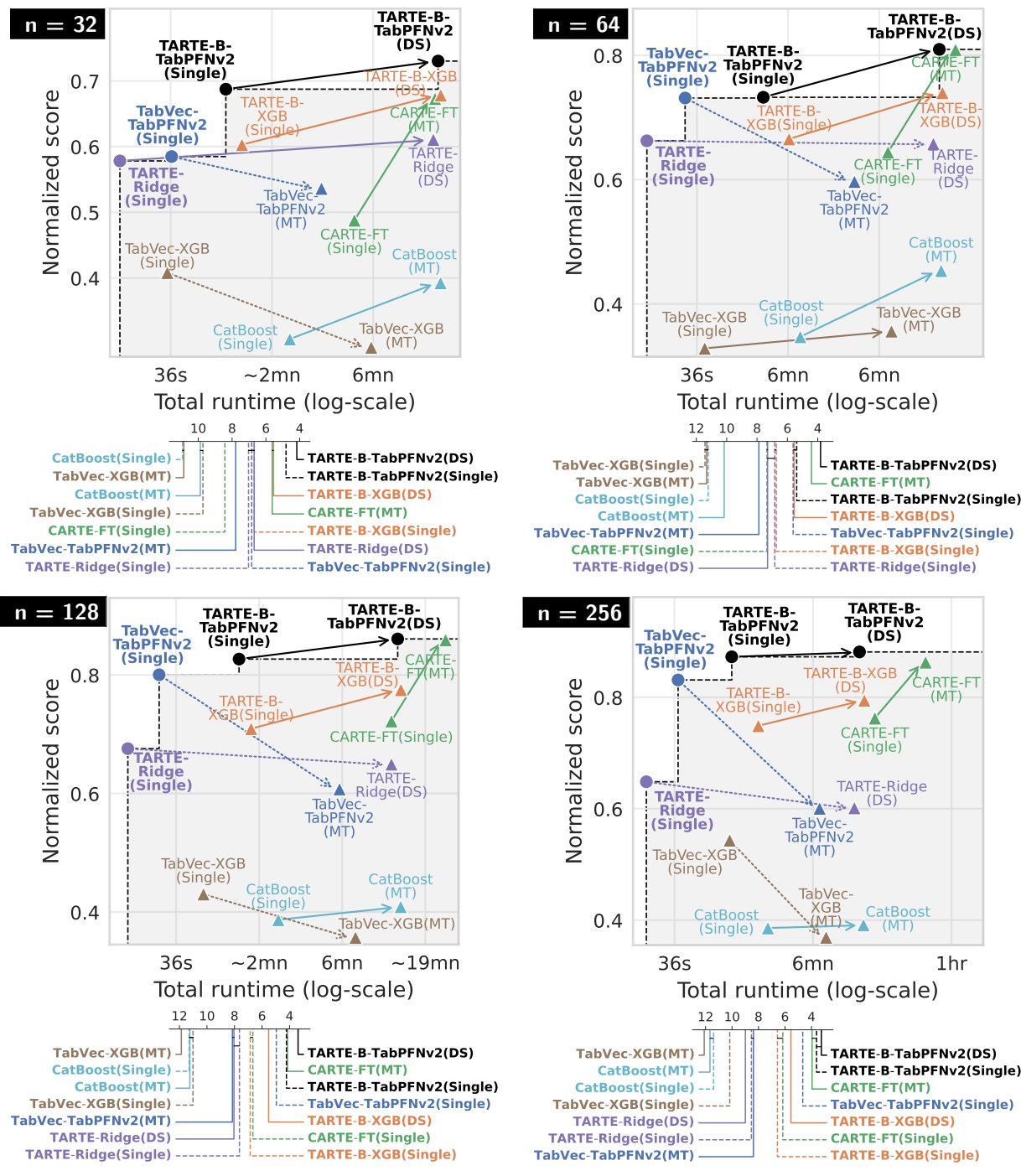

Figure 9: **Results for domain specialization with single source table** Regardless of the train-sizes, TARTE–B can provide benefits with domain specialization. Once domain specialized models are available, TARTE can readily benefit without requiring complex refitting with the source tables.

