# OpenReview forum: "Table Foundation Models: on knowledge pre-training for tabular learning"
_TMLR — Accepted by TMLR_

### Review · Reviewer_FSQi · 2025-06-03

**Summary Of Contributions:**

This paper introduced TARTE, a foundational model for tabular data. The primary highlight is knowledge pre-training to generate reusable vector representations of table rows. A prior model CARTE serves as a primary baseline, but the new model allows generalizable and reusable baselines reducing the need for domain specific fine tuning. The architecture is a transformer-based model at its core, which jointly processes column names and cell values, enabling context-aware representations. supporting several downstream paradigms (e.g., featurization, boosting, domain transfer). Empirical evaluations show that TARTE outperforms tree-based and neural baselines including TabPFNv2 and CARTE above, especially in few-shot and domain-transfer scenarios, while also offering favorable compute-performance trade-offs.

**Audience:**

Yes

**Claims And Evidence:**

Yes

**Requested Changes:**

* It is my understanding that CARTE and TARTE have different pre-training datasets. It seems possible to me that the differences in performance entirely stem from this fact. This should also be mentioned in Section "3.4 Differences to the CARTE approach", as it seems like a key difference.
* On a similar note, it might be beneficial to include some ablation studies on architecture or preprocessing steps, as far as computational resources allow. This will certainly strengthen the paper.
* How were the source and target tables selected/determined?
* On page 2, "we first showing how" -> "we first show how"

**Strengths And Weaknesses:**

### Strengths
Overall, the paper is well written quite easy to follow. I appreciate the authors in drafting a well written manuscript! Some key strengths:
* TARTE effectively bridges the gap between tabular learning and foundation models by incorporating data semantics through pre-training.
* Strong empirical validation across diverse datasets (classification and regression, varying sizes and data types).
* Multiple post-training options (fine-tuning, feature extraction, boosting) make TARTE adaptable to diverse use cases

### Weaknesses
* A majority of the architectural novelty comes from the baseline model CARTE, so the paper is more about how to modify CARTE to better world considerations like efficient computation and better latency.
* See Requested changes

---

> ### Author Response · Authors · 2025-06-29
> **Response to Reviewer FSQi**
>
> We appreciate the valuable comments of Reviewer FSQi which were helpful to improve the quality of the paper. Find below the responses or the changes to the manuscript.
>
> **Differences to CARTE and ablation on components of TARTE:**
> As outlined in subsection 3.4, TARTE deviates from CARTE in several ways, and certainly the pre-train data is one of them. To observe the effect of the different factors, we compared various pre-training schemes concerning TARTE and CARTE in subsection 4.3. The review rightfully points out that these comparisons could be improved, for instance adding results to check if the differences in performance entirely stem from the pre-train data. To enrich our analysis, we have conducted (1) CARTE pre-training schemes on the TARTE pre-train data, and (2) TARTE pre-training schemes on CARTE architecture.
>
> The updated Figure 6 shows that CARTE pre-training scheme, regardless of pre-train data, actually hurts the downstream prediction, while changing the pre-training scheme to TARTE with CARTE architecture helps. However, the architectural and preprocessing components of TARTE are needed to further improve the knowledge pre-training. In Figure 6, we additionally include results that exclude column information (architectural) and datetime detection (preprocessing) components. The results describe that each component plays a crucial role for improvements.
>
> **(Q1) How were the source and target tables selected/determined?**
> We thank the reviewer for suggesting clarification improvements to the paper. The original manuscript failed to describe well how the source-target tables are selected/determined. For domain specialization, we find datasets from a similar domain (e.g., wine, movies, restaurants). The tables are within the same domain, but acquired in different settings. For the experiments, we consider all possible source(s)-target cases for a specific domain of interest. We have included a clarification in the revised manuscript (subsection 4.4 and appendix B.3).
>
> **(Q2) On page 2, "we first showing how" -> "we first show how":**
> We thank the reviewer for taking a close look for grammatical errors in the manuscript. We have fixed those errors. We will continue to proof-read the manuscript.

---

### Review · Reviewer_iJVz · 2025-06-09

**Summary Of Contributions:**

This paper proposes a new pre-training scheme named TARTE for tabular data. TARTE models each row in a table as a pair containing the embeddings of the column name and the cell value. It is pre-trained on large knowledge bases using rational data and is able to benefit downstream tasks through several different approaches, including finetuning, serving as a table featurizer, and boosting another complementary model.

**Audience:**

Yes

**Claims And Evidence:**

Yes

**Requested Changes:**

Requested changes:
1.	Please give a brief introduction to what a critical difference diagram is and what message it conveys. This would make the experiment section more self-contained

For other requested changes, please refer to the Weaknesses section.

**Strengths And Weaknesses:**

Strengths:

1.	The paper has clear presentation and is easy to follow.

2.	The paper conducts abundant experiments and analyzes in which scenarios TARTE has more gains than other methods. This gives a comprehensive picture of the capability boundary of TARTE.

3.	The proposed method achieves a good balance between accuracy and efficiency.

Weaknesses:

1.	In the paragraph “TARTE as a table featurizer,” the paper claims that “TARTE can be used to generate meaningful embeddings for table entries.” However, there are no further details on how to generate those embeddings. It is suggested that the authors elaborate on this part.

2.	It is also suggested that the authors provide more detailed descriptions on how “boosting a complementary model” is done.

3.	In Fig. 3, it seems that boosting with TARTE consistently outperforms other methods (including those variants of TARTE). However, there are no baseline boosting methods using other pretrained tabular models (e.g., CARTE). The authors are suggested to explain this issue. If boosting with other pretrained tabular models is feasible, these results should be included in the comparison. If not, please clarify the reason.

4.	It is not explained why in Fig. 3 the boosting variant TARTE-B-TabPFNv2 consistently outperforms the finetuning variant TARTE-FT. The authors are encouraged to explain where this performance gain comes from, and when boosting tends to perform worse than finetuning.

---

> ### Author Response · Authors · 2025-06-29
> **Response to Reviewer iJVz**
>
> We appreciate the valuable comments of Reviewer iJVz which were helpful to improve the quality of the paper. Find below the responses or the changes to the manuscript.
>
> **Comparing baseline of TARTE-B:**
> We agree that in the original manuscript, there was no baseline on the boosting methods using other pretrained tabular models. As a follow-up experiment, we applied the boosting used for CARTE embeddings (denoted as CARTE-B-TabPFNv2). The results show that CARTE-B-TabPFNv2 does not always bring marked benefits compared to the base model (TabVec-TabPFNv2), highlighting that TARTE boostings can complement the base model through better pre-training. The results of CARTE-B are included in subsection 4.2 and appendix D.1.
>
> **Performance difference between TARTE-B-TabPFNv2 and TARTE-FT:**
> We thank the reviewer for pointing out the clarity of the results. From the experiments, we see that boosting with TARTE embeddings give positive impacts for prediction; however, it requires a strong base tabular model to achieve the best performances as shown with the difference between TabPFNv2 and XGB. On the other hand, TARTE-FT does not require a base model, but it requires extensive hyperparameter tuning. Due to the computational drawbacks, the search space for TARTE-FT is rather limited. Thus, the performance gap between TARTE-B-TabPFNv2 and TARTE-FT may be due to lack of tuning for fine-tuning. We have included the new insights on the results in the revised manuscript (subsection 4.2).
>
> **Elaboration of TARTE variants:**
> Thank you for suggesting clarifications and elaborations on TARTE variants for downstream learning, which are crucial elements of TARTE.
>
> The TARTE featurizer works similarly to the SentenceTransformers, in which the processed input is passed through the frozen backbone. After passing through the layers, TARTE featurizer extracts the readout element T (similar to the [CLS] token in LLMs), which  can be used in any machine model as a pipeline.
>
> The TARTE boosting (denoted as TARTE-B) takes extracted embeddings further: From the base tabular model trained with the  original input, we additionally train a model by fitting the train residuals of the base model with TARTE embedded features from TARTE featurizer. The predictions are made by simply adding the outputs of the two models, which can be considered as a boosting approach.
>
> To reflect the comment, we have updated the explanation of the TARTE variant (subsection 3.3) in the revised manuscript.
>
> **Information of critical difference diagram:**
> Thank you for pointing out the missing information on the critical difference diagram. We have included a description and the message it conveys where it is first introduced in the revised manuscript (subsection 4.2, in particular footnote 4).

---

### Review · Reviewer_GrFF · 2025-06-13

**Summary Of Contributions:**

This paper introduces TARTE (Transformer Augmented Representation of Table Entries), a pretrained model for tabular data that leverages knowledge pre-training to learn effective representations of both string and numerical entries in tables. Unlike prior tabular models that mainly benefit from fine-tuning, TARTE demonstrates that pretraining alone can yield strong, reusable representations for a wide range of downstream tasks. The key contributions are:
1. A novel Transformer-based architecture for tabular data encodes cell values enriched with column names, leveraging large-scale factual and numerical data during pre-training. This enables the model to capture semantic associations between entries without requiring access to original pre-training tables during downstream tasks, supporting both fine-tuning and frozen usage in diverse downstream applications.
2. Extensive empirical validation over small- and mid-sized tables, demonstrating that TARTE consistently outperforms state-of-the-art baselines (e.g., tree-based and neural models) in both accuracy and efficiency. Detailed analysis of pretraining effectiveness, showing that TARTE performs best when downstream tables contain rich and diverse string-based features that overlap with the pretraining corpus.
3. Demonstration of TARTE's transferability, highlighting its ability to specialize to new domains via domain-specific finetuning or feature reuse, in line with foundation model paradigms.

**Audience:**

Yes

**Claims And Evidence:**

Yes

**Requested Changes:**

Please see the weakness part.

The authors may also consider larger-scale datasets, and include the results of some non-pretrain methods such as FT-T, ModernNCA, RealMLP, and TabR.

**Strengths And Weaknesses:**

Strengths:

1. TARTE demonstrates that knowledge pre-training can directly improve downstream performance. This contrasts with most existing methods that require fine-tuning.
2. The model supports multiple deployment modes allowing flexible trade-offs between accuracy and computational cost.
3. The paper shows that TARTE achieves Pareto-optimal performance, balancing runtime with predictive quality. It is also shown to be reusable, saving training cost in multi-task or transfer settings.
4. The authors provide an in-depth study of factors influencing pretraining effectiveness, including string similarity, column types, and data heterogeneity. This adds clarity to the model's practical applicability.

Weaknesses:
1. The FastText-based encoding has known limitations with long or complex string entries, and TARTE inherits this weakness.
2. The effectiveness of TARTE may diminish on domains with unfamiliar terminology or structure.

---

> ### Author Response · Authors · 2025-06-29
> **Response to Reviewer GrFF**
>
> We appreciate the valuable comments of Reviewer GrFF which were helpful to improve the quality of the paper. Find below the responses or the changes to the manuscript.
>
> **Limitations of FastText embeddings:**
> As the reviewer has pointed out, FastText embedding holds limitations for long string entries, and we agree that TARTE inherits this weakness (as noted in the last paragraph of subsection 4.3). We selected FastText since string (text) entries in tables mostly contain only a few words (across 51 datasets, the median ratio of unique entries with more than 10 words is 0.025). While the choice of language models for semantic pre-training remains subjective, we agree that an interesting alley to explore would be using different language models depending on different structure of strings (e.g., simple words or sentences). Such research requires using more costly language models in the pretraining runs of the table foundation model, such as TARTE. As a consequence, it is a significant computational effort, and we did not have time to carry it out for the revision. Rather, in the revised manuscript, we have added a related discussion (appendix A.2) and improved the caption of table 1 to point out this phenomena.
>
> **Effectiveness of TARTE may diminish on domains with unfamiliar terminology:**
> We agree with the review that TARTE will be most useful on domains with terminology related to the pretraining data. We have highlighted this in the manuscript (subsection 4.3), showing that the quality of the pre-train data to reflect more of the downstream tasks is important. In particular, the importance of the “average inlier probability” in table 1 reveals exactly the phenomena that the reviewer has in mind. We have improved the caption of this table to point this out. In the long term, we believe that the challenge can be overcomed by enlarging the spectrum of  pre-train datasets or by fine-tuning to specialize to a specific domain.
>
> **Larger-scale datasets and additional baselines:**
> For larger-scale datasets, we are nearing the maximum capacity of TabPFNv2 and it requires architectures that scale better. An option would be to use the related TabICL [1], which scales better. However, TabICL does not currently support regression tasks. For our future work, we hope to benefit from improved scalability, and thus to study also larger datasets.
> Concerning additional baselines, we included a non-pretrain baseline of RealMLP [2] (subsection 4.2, both on small tables and 10k experiments). The new results show that while RealMLP can achieve strong performances as a non-pretrain baseline, TARTE continues to outperform, highlighting the importance of background information from pre-training.
>
> [1] Qu, Jingang, et al. "TabICL: A Tabular Foundation Model for In-Context Learning on Large Data." arXiv preprint arXiv:2502.05564 (2025).
>
> [2] Holzmüller, David, Léo Grinsztajn, and Ingo Steinwart. "Better by default: Strong pre-tuned mlps and boosted trees on tabular data." Advances in Neural Information Processing Systems 37 (2024): 26577-26658.

---

### Decision · Action_Editor_UT2U · 2025-07-21

**Recommendation:** Accept with minor revision

**Additional Comments:**

For Figure 2, the YAGO project (https://yago-knowledge.org/) is licensed under CC BY 4.0 (https://creativecommons.org/licenses/by/4.0/). Therefore,  the credit of the YAGO project should be added to cite its work. Adding credits for Wikidata is not mandatory as it is made public under CC0 (https://creativecommons.org/public-domain/cc0/). Of course, it does not prevent the authors from adding Wikidata’s credit.

**Audience:**

Yes

**Audience Explanation:**

The development of foundation models for tabular data has been one of the main topics of machine learning research in recent years. This paper proposes a new foundation model that achieves good performance. Additionally, the reviewers unanimously agreed that it met these criteria. Therefore, this paper is of interest to audiences who work in this area.

**Claims And Evidence:**

Yes

**Claims Explanation:**

This paper proposes TARTE, a foundation model for table data. The main claims of this paper are as follows:

- TARTE works as a featurizer for tabular data that can be applied to downstream tasks with low learning costs.
- By combining fine-tuning and boosting, TARTE achieves Pareto-optimal in terms of computational cost and prediction performance.
- Fine-tuned TARTE enables transfer learning for domain-specific tasks.

The paper was reviewed by three expert reviewers, who unanimously agreed that it meets the acceptance criteria of TMLR. Also, the authors responded appropriately to the reviewers' questions and revised the paper as necessary.

Based on the reviewers' comments and the revisions, I have determined that the above claims are supported by appropriate evidence.